# R2DT is a framework for predicting and visualising RNA secondary structure using templates

Blake A. Sweeney[1,7], David Hoksza [2,7], Eric P. Nawrocki[3], Carlos Eduardo Ribas [1], Fábio Madeira [1], Jamie J. Cannone[4], Robin Gutell[4], Aparna Maddala[5], Caeden D. Meade[5], Loren Dean Williams [5], Anton S. Petrov [5], Patricia P. Chan [6], Todd M. Lowe[6], Robert D. Finn [1,8] & Anton I. Petrov [1,8 ✉]

Non-coding RNAs (ncRNA) are essential for all life, and their functions often depend on their secondary (2D) and tertiary structure. Despite the abundance of software for the visualisation of ncRNAs, few automatically generate consistent and recognisable 2D layouts, which makes it challenging for users to construct, compare and analyse structures. Here, we present R2DT, a method for predicting and visualising a wide range of RNA structures in standardised layouts. R2DT is based on a library of 3,647 templates representing the majority of known structured RNAs. R2DT has been applied to ncRNA sequences from the RNAcentral database and produced >13 million diagrams, creating the world's largest RNA 2D structure dataset. The software is amenable to community expansion, and is freely available at https://github.com/rnacentral/R2DT and a web server is found at https://rnacentral.org/r2dt.

[1] European Molecular Biology Laboratory, European Bioinformatics Institute, Cambridge, UK. [2] Department of Software Engineering, Faculty of Mathematics and Physics, Charles University, Prague, Czech Republic. [3] National Center for Biotechnology Information, National Library of Medicine, National Institutes of Health, Bethesda, MD, USA. [4] Department of Integrative Biology, The University of Texas at Austin, Austin, TX, USA. [5] School of Chemistry and Biochemistry, Center for the Origins of Life, Georgia Institute of Technology, Atlanta, GA, USA. [6] Department of Biomolecular Engineering, University of California Santa Cruz, Santa Cruz, CA, USA. [7] These authors contributed equally: Blake A. Sweeney, David Hoksza. [8] These authors jointly supervised this work: Robert D. Finn, Anton I. Petrov. ✉email: apetrov@ebi.ac.uk

RNA molecules are key components of a wide range of biological processes, such as translation, splicing, and transcription. For many RNAs, the 3D structure is essential for biological function. For example, ribosomal RNA (rRNA) and transfer RNA (tRNA) adopt very specific, evolutionarily conserved 3D conformations in order to perform translation, and RNA aptamers can specifically recognise small molecules and other ligands by virtue of their 3D structures. The architecture of a structured RNA molecule is hierarchical, whereby the RNA sequence (primary structure) folds into local elements that, in turn, interact with each other to form the 3D structure[1]. The majority of intramolecular contacts in most ncRNAs can be represented in the form of 2D structure diagrams, which are far more accessible and can present a broader variety of information than the corresponding 3D structures.

Many RNAs are visualised following standard, community-accepted conventions. For example, the 2D diagrams from the Comparative RNA Web Site[2] (CRW) have been used for decades and are widely accepted as standard for rRNA visualisation. Similarly, tRNAs are traditionally displayed in a cloverleaf layout with the 5′- and 3′- ends located at the top, the anticodon loop pointing towards the bottom, and the D- and T- loops facing left and right, respectively[3]. Both of these representations capture important structural and functional features, providing valuable insights into the RNA structure and function. However, the construction of RNA secondary structures requires manual curation, which does not scale with the large numbers of sequences being generated by modern molecular biology techniques.

There are many automated approaches for visualising RNA 2D structure that can be broadly categorised into one or more of the following groups based upon the type of visualisation and the algorithm used to generate it. Visualisations may (1) Use dot plots where the $X$ and $Y$ axes represent the RNA sequence and the base pairs are shown as dots in Cartesian space (for example iDotter[4]); or (2) Arc plots[5] which represent the RNA as a line and interactions as arcs connecting paired nucleotides (for example R-Chie[6]); and finally (3) Circular diagrams, which are effectively arc diagrams folded into a circle (for example VARNA[7]). Algorithmically, the approaches may use (1) Force-directed layouts where nucleotides represent the nodes of a graph and base pairs as well as other interactions are shown as lines connecting the nodes. These layouts are governed by the attraction and repulsion between the nucleotides and interactions (for example Forna[8]); or (2) Rule-based methods[9], such as RNAView[10], 3DNA[11], PseudoViewer[12], R2R[13], RNA2Drawer[14], jViz[15], RNApuzzler[16] and many others, for a more comprehensive review see[17,18]. However, none of these methods achieves all five properties of ideal RNA 2D structure diagrams[18,19]: they should be modular to reflect functional domains, appear similar for related structures despite sequence or minor structural changes to allow for easy comparison, avoid overlaps and ensure visual clarity, be a realistic representation of the 3D structure (if known), and finally be aesthetically pleasing[18,19].

In general, existing tools do not produce diagrams in widely accepted layouts, and in some cases, homologous or even identical sequences are displayed in different layouts that can be hard to analyse and compare[17]. Specifically, while dot plots are useful for comparative analysis in the hands of expert users[20], they are difficult for users to interpret; arc plots, which show sequence conservation and base pairs, can be prohibitively large and make it difficult to examine small structural changes; circular diagrams can be uninformative and can make small structural changes hard to analyse; while force-directed layouts are not always robust against changes in the sequence or structure, making a comparison across related structures difficult. The rule-based methods can reliably produce diagrams even for large RNA structures[16],

but they do not take advantage of the community-accepted layouts. In addition, some methods can show other issues such as overlaps and a lack of visual clarity (Fig. 1). Some packages, like the SSU-ALIGN[21] can generate 2D structure diagrams of SSU rRNA following the CRW layout but display only a fixed number of consensus positions.

The lack of tools for visualising RNAs in consistent, reproducible, and recognisable layouts, makes comparing RNA structures difficult for RNA biologists and essentially impossible for non-specialists.

Here we fill a fundamental gap in visualising structured RNAs by introducing R2DT (RNA 2D Templates). R2DT encapsulates a comprehensive pipeline for template-based RNA 2D visualisation, generating diagramatic 2D representations of RNA structures based on a representative library of templates, and is implemented as both a standalone application (https://github.com/rnacentral/R2DT) and a web server (https://rnacentral.org/r2dt). The framework can be easily updated and extended with additional templates, and it has been extensively tested on > 13 million sequences from RNAcentral[22], a comprehensive database of ncRNA sequences (see Validation for more information).

## Results

**Automatic pipeline for template selection and 2D structure visualisation.** We developed a computational pipeline that uses a template library to define standard layouts for different types of RNA. A minimal template contains a reference sequence, as well as cartesian coordinates for each nucleotide, and a 2D representation of the structure in dot-bracket notation that encapsulates the canonical Watson-Crick base pairs. Some templates also contain the wobble GU base pairs, but non-canonical base pairs are not currently included in the templates (see the next section for the detailed description of the template library).

To enable automatic template selection, for each template a covariance model is generated using Infernal[23] based on the reference sequence and its 2D structure. The R2DT pipeline includes the following steps:

1. For each input sequence, the top scoring covariance model is selected using the *ribotyper.pl* programme in the Ribovore software package (version 0.40) (https://github.com/ncbi/ribovore). For model selection, *ribotyper.pl* runs the Infernal[23] cmsearch programme and uses a profile HMM derived from the covariance model that scores sequence only and ignores secondary structure to limit running time. If Ribovore does not find any matches, tRNAscan-SE 2.0[24] is used to search query sequences against the tRNA models. To speed up template selection, the library is divided into several subsets which are processed separately (Rfam, LSU and SSU RiboVision rRNAs, CRW, and tRNA templates). If a sequence is classified to a template model in one of the subsets (defined as being designated "PASS" by *ribotyper.pl* without a "MultipleHits" flag) then the remaining subsets are not searched. In cases where both a 3D-based and a covariation-based template are available for the same RNA, the 3D-based template is preferentially selected. The Ribovore software is used to search against all models except for tRNA. If no hits are detected, tRNAscan-SE 2.0 is then used to compare the sequences against the bacterial, archaeal, and eukaryotic domain-specific tRNA models. Once a top scoring domain-specific tRNA model is chosen, the sequence is compared with the isotype-specific tRNA models for that domain. It is also possible to specify a template and bypass the classification step. If the sequence

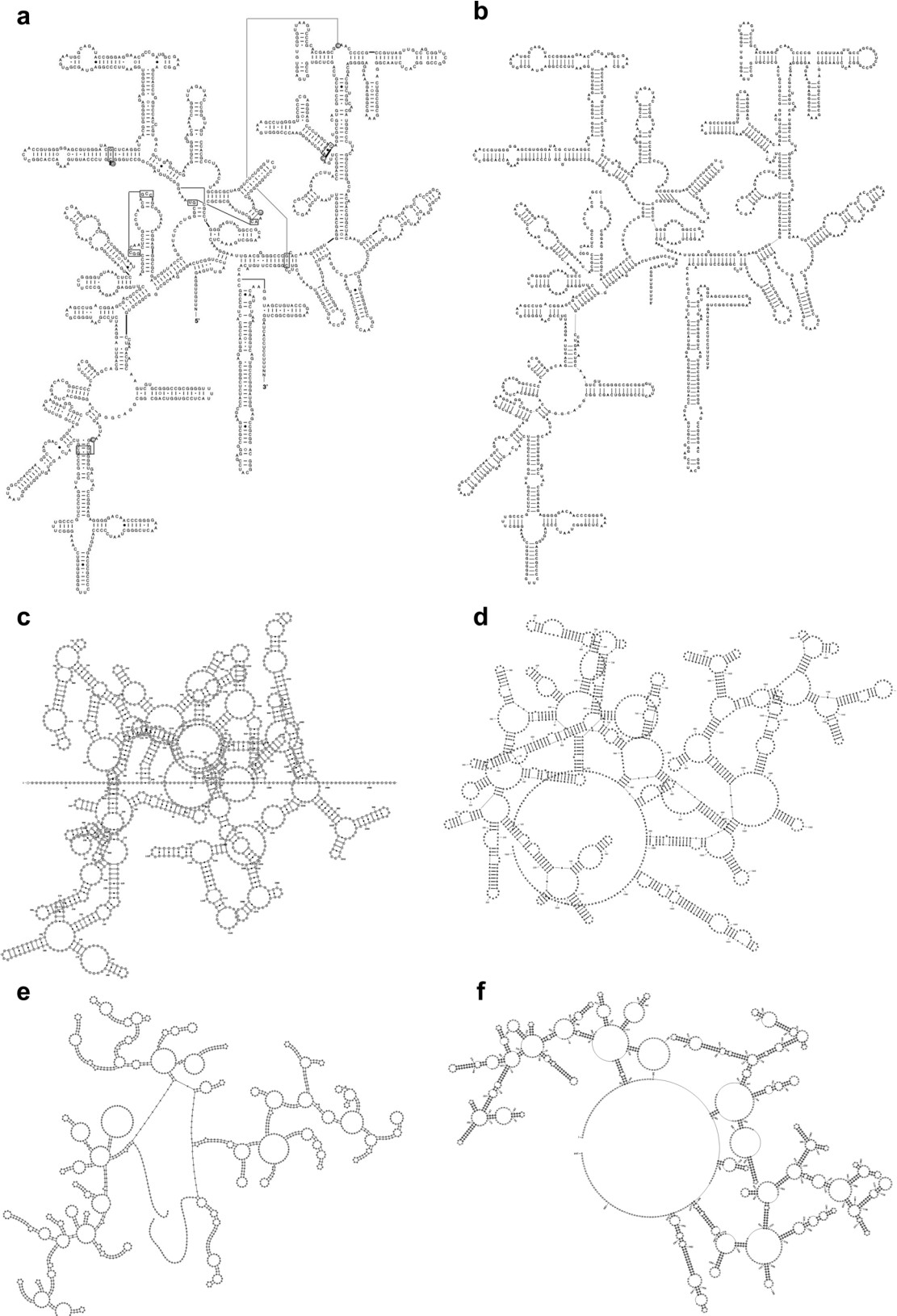

**Fig. 1 Examples of 2D structures of the *Thermus thermophilus* SSU rRNA. a** A manually curated 2D structure from CRW[2]; 2D structures from **b** R2DT using the layout from diagram a as a template; **c** Varna[7]; **d** RNA2Drawer[14]; **e** Forna[8]; **f** PseudoViewer[12]. Diagrams **b–f** share the same sequence (RNAcentral accession URS000080E226_274) and 2D structure; however, only diagram **b** reflects the SSU topography.

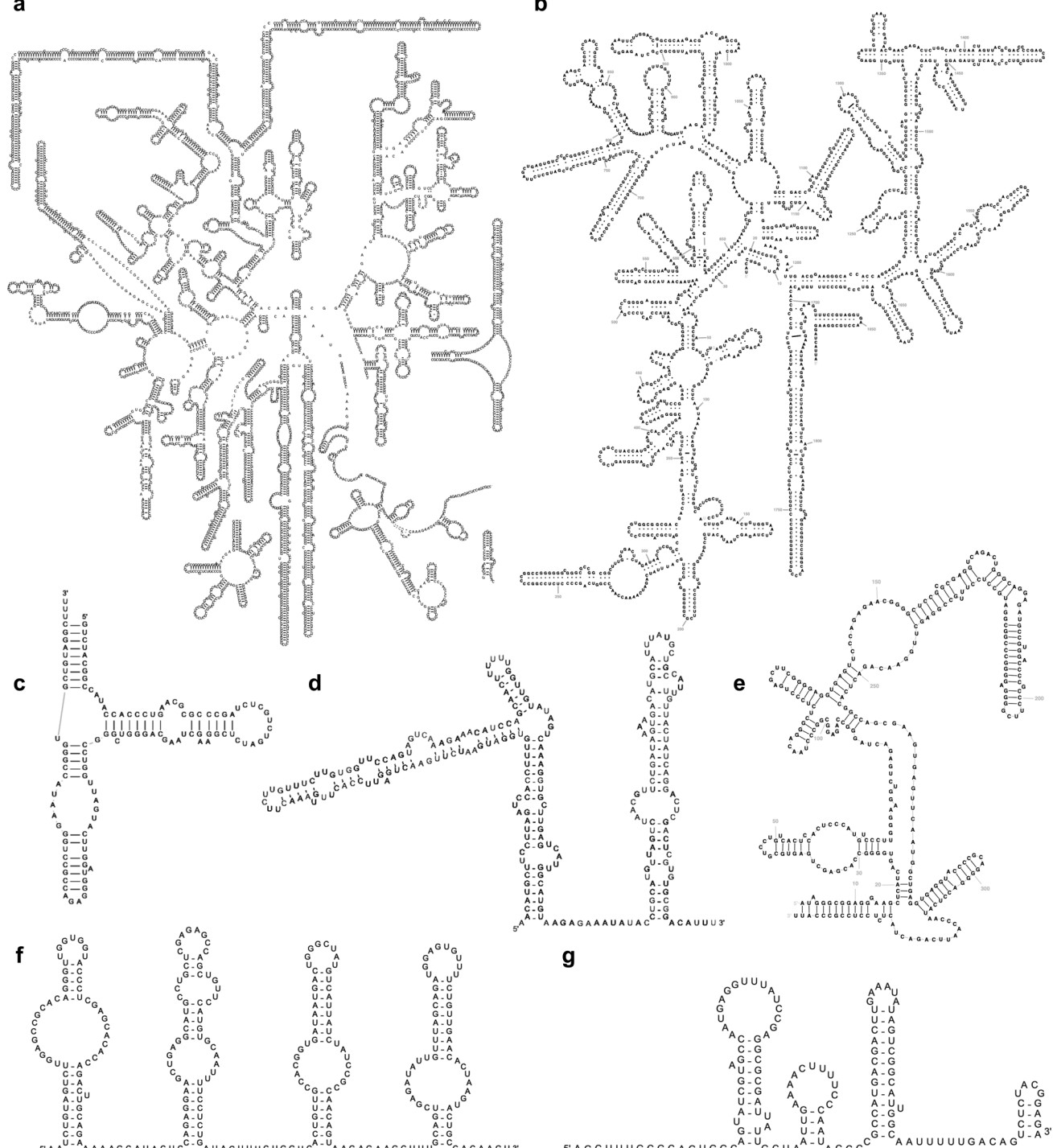

**Fig. 2 Example human RNA 2D structures generated by R2DT. a** Cytoplasmic LSU rRNA (RNAcentral accession URS0000ABD82A_9606);
**b** cytoplasmic SSU rRNA (URS0000726FAB_9606); **c** 5S rRNA (URS00000F9D45_9606); **d** SNORA53 RNA (URS000075B5A7_9606); **e** RNAse P
(URS000013F331_9606); **f** SCARNA13 RNA (URS000026BDF0_9606); **g** U4 snRNA (URS0000149178_9606).

does not match any templates, the following steps are skipped and no output files are generated.

2. The input sequence is folded with the Infernal cmalign programme using the top scoring covariance model. This ensures that the predicted 2D structure is compatible with the template 2D structure. It is important to note that R2DT does not attempt to fold the unstructured regions found in some templates or predict the structure of the insertions relative to the template.

3. The 2D structure and the selected template are used by the Traveler software[25] to generate a 2D structure diagram (see examples in Fig. 2).

The 2D structure of the input sequence is predicted using Infernal based on the template covariance model, so the template serves both as a source of coordinates for nucleotides when positioned on the diagram and a source of base pairing information. The input sequence is not required to closely match the template, as insertions and deletions can be accommodated,

and nucleotides can be repositioned depending on the structural context by the Traveler software[25].

For each sequence, the pipeline produces a text file with the 2D structure in dot-bracket notation and a 2D diagram in SVG format. The diagrams are coloured depending on the identity of the individual nucleotides in the input sequence relative to the template. Identical nucleotides are shown in black, while inserted nucleotides are displayed in magenta. If a nucleotide is modified compared to the template reference sequence, it is shown in green. If the location of the nucleotides was automatically repositioned relative to its corresponding position in the template, the nucleotide is coloured blue.

The SVG diagrams can be scaled to any resolution and edited using text editors or specialised vector graphics editing software. When viewed with a web browser, additional information is shown when hovering the mouse over individual nucleotides (for example, hovering over modified nucleotides reveals the identity of the nucleotide in the corresponding position of the reference sequence). Further interactivity can be added to the SVG visualisations using JavaScript and CSS web technologies.

**Comprehensive 2D structure template library**. We compiled a library of 3647 templates aggregating RNA 2D structure layouts from different sources (Table 1) in order to represent the diversity of RNA structures ranging from <100 nucleotides (tRNA) to >5000 nucleotides (human large subunit rRNA). Templates can be annotated with additional metadata about the RNAs, such as a taxonomic distribution or subcellular localisation, as well as per-nucleotide annotations that can be transferred to the corresponding nucleotides of the input sequence (for example, tRNA nucleotide numbering using the Sprinzl scheme[26]).

While the majority of the 3647 templates were integrated from the existing sources (Table 1), 103 templates have been manually curated specifically for this project, as described below (also see Supplementary Table 1).

The availability of the experimentally determined ribosomal 3D structures enabled us to improve the traditional rRNA diagrams available from the CRW[2,27]. Specifically, the 3D structural data assessed the accuracy of the covariation-based 16S and 23S rRNA secondary structures, removed the few incorrect base pairs, added missing base pairs with both Watson-Crick and non-canonical base pair conformations, and provided detailed modelling of the species-specific expansion segments that were not present in the covariation-based expansion segments. The revised LSU 2D templates are outlined using single-page layouts and explicitly depict H26a[28], a helix that connects the 5′ and 3′ halves of the LSU rRNA. This irregular helix, which is now known to be the loop-E motif[29] was initially suggested by Haselman and Fox[30], and had

been indicated by arrows connecting the two halves of the historical LSU rRNA layouts[31]. All non-canonical interactions were explicitly depicted when the first 3D structural model of the LSU particle became available[32]. The single-page LSU layouts enable R2DT to visualise the LSU 2D structures in standard orientations completely automatically, which has not been possible until now (Fig. 3a). For the SSU rRNA, the updated 2D structures use a more accurate representation of the central pseudoknot, reflecting the existence of the base triplexes. In addition, the 3D structures allowed us to visualise the structure of the species-specific eukaryotic expansions[33,34] that could not be modelled using covariation analysis alone (Fig. 3b).

The resulting rRNA structures are up-to-date, consistent with the 3D structures, and broadly sample the phylogenetic tree (the templates are listed in Supplementary Table 1). Both LSU and SSU layouts are generalisable to accommodate numerous expansions that exist in eukaryotic species.

Although cytosolic tRNAs are generally known to have a cloverleaf 2D structure, different isotypes (the tRNA families inserting different amino acids) have distinct "identity elements" recognised by specific aminoacyl tRNA synthetases for charging the tRNAs with the proper amino acids. In addition to the tRNA anticodon that binds with the mRNA codon during translation, these identity elements include discriminatory nucleotides and base pairs throughout the tRNA sequences and vary across the domains of life[35]. To better represent the tRNA structures, we prepared 68 isotype-specific templates for bacterial, archaeal, and eukaryotic tRNAs that include those decoding the standard twenty amino acids, initiator methionine/N-formylmethionine (tRNA$^{iMet}$ in archaea/eukaryotes or tRNA$^{fMet}$ in bacteria), isoleucine for the AUA codon in bacteria and archaea, and selenocysteine (Fig. 4). Consensus tRNA primary sequence with the 2D structure for each isotype of each taxonomic domain was generated based on the tRNA alignments used for building the isotype-specific covariance models in tRNAscan-SE 2.0[24]. The isotype-specific tRNA 2D structure templates were created using the corresponding consensus sequences and structures. In addition, we generated six domain-specific templates for more general application. Due to the structural difference of the variable loop in type I and type II tRNAs[36], alignments for building the domain-specific covariance models in tRNAscan-SE 2.0[24] were divided into two sets. Similar to the isotype-specific ones, the domain-specific templates were built with the consensus sequences and structures for both type categories of tRNAs. Together, the isotype-specific templates can be used to visualise 2D structures of tRNAs with typical features while the domain-specific templates can be applied for the atypical predictions with undetermined or inconsistent isotypes.

The R2DT pipeline is designed to be extendable as templates are added to the library. Notably, R2DT can also serve as a tool for the development of templates where the R2DT output is used as a starting point for manual refinement of the 2D layouts. To facilitate the workflow, we provide a modified version of the XRNA software[37] called XRNA-GT that supports the import of the R2DT-generated SVG files and can be used to adjust the 2D layouts (for example, change the orientation of RNA helices or edit base pairs). Using XRNA-GT it is also possible to add custom annotations, such as helix or nucleotide numbers, in order to produce publication-ready images. The updated 2D layouts can be submitted to the R2DT library where they become templates, upon review by the R2DT team. This workflow has been successfully used internally to produce the 3D-based SSU templates. In addition, users can apply their own structures to create templates starting with the FASTA and XML files defining the sequence, secondary structure, and the $X, Y$ coordinates of each nucleotide (example input files and a script for template

**Table 1 The RNA 2D structure template library (* manually curated templates developed specifically for this project).**

| RNA type | Template source | Number of templates | Manually curated? |
|---|---|---|---|
| SSU rRNA | CRW (covariation-based) | 654 | Yes |
| | RiboVision (3D-based) | 8* | Yes |
| LSU rRNA | RiboVision | 21* | Yes |
| 5S rRNA | CRW | 200 | Yes |
| tRNA | GtRNAdb | 74* | Yes |
| RNAse P | RNAse P database | 17 | Yes |
| | | 2* | Yes |
| Small RNAs | Rfam | 2671 | No |
| | | Total: 3647 | |

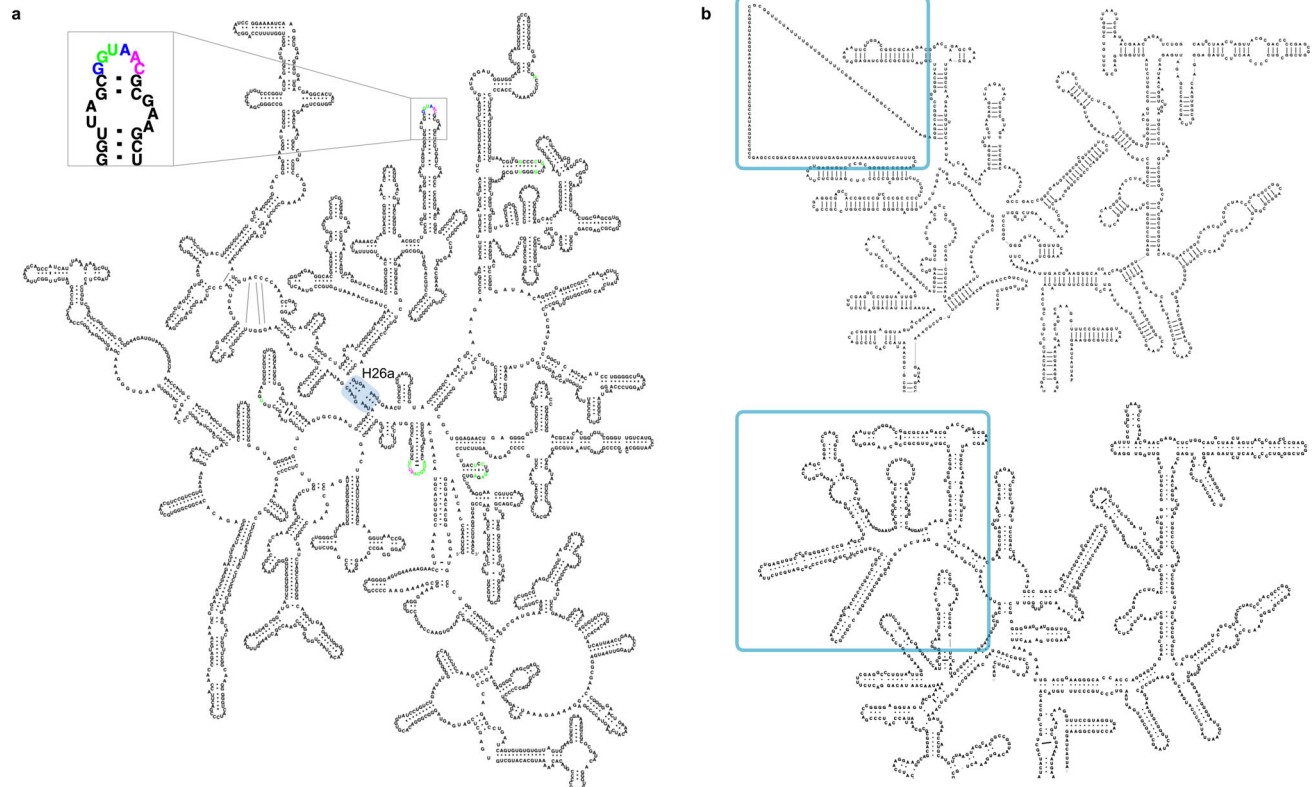

**Fig. 3 Example of 3D structure-based rRNA templates. a** An *Escherichia coli* LSU rRNA is displayed by R2DT using a single-page layout (URS0000051AF4_562). Helix 26a is highlighted with a blue box. An inset shows a zoomed-in fragment with nucleotides that are identical between the template and the sequence shown in black, insertions shown in magenta, and nucleotides that are different between the template and the sequence shown in green. **b** A fragment of a covariation-based human SSU rRNA layout based on the CRW template (top) and the revised, 3D structure-based template showing additional base-pairing interactions (bottom). The species-specific region is highlighted in blue.

generation are included in the documentation). We welcome contributions from the community and provide detailed documentation on GitHub (https://github.com/RNAcentral/R2DT#how-to-add-new-templates).

**Validation of 2D diagrams**. At the time of writing, there are no alternative methods that enable template-based RNA 2D structure visualisation at a comparable scale. The only related method, implemented in rPredictorDB[38], has a small number of templates (56 as of July 2020) and a limited support for alternative templates from the same RNA type (for example, species-specific rRNA templates). As this is a unique dataset, we developed global benchmarks to assess both accuracy of the template selection and the quality of the resulting 2D diagrams.

We tested R2DT with a diverse set of rRNA sequences to evaluate the template selection process, focusing on the rRNA templates as they are annotated at the species level, making it possible to compare the taxonomic lineages of the input sequence and the template. We selected all rRNA sequences from RefSeq[39] shorter than 10,000 nucleotides (23,843 sequences as of July 2020). The sequences were visualised with R2DT and the taxonomic trees of the sequences and the selected templates were compared by identifying the most specific taxonomic rank common to the templates and the RefSeq sequences. For example, if an rRNA from *Photorhabdus caribbeanensis* was drawn using a template from *Escherichia coli*, their respective phylogenies share the order *Enterobacteriales*, thus the sequence and the template agree at the level of order. The majority of sequences match templates at the level of kingdom (55.5%), phylum (20.0%), or class (16.1%) (Supplementary Table 2), indicating that the

selected templates can be taxonomically distant from the input sequences. This effect is due to the preferential use of the 3D-based SSU and LSU rRNA templates, as only a relatively small number of 3D structures is available. However, when we classified each nucleotide in the 2D diagrams based on whether it matched a template for each taxonomic rank separately, we found that at least 94% of all nucleotides were in the same position as the template for all taxonomic ranks, confirming that the sequences closely matched the selected templates despite the phylogenetic distance between the template and sequence.

We evaluated R2DT performance on a set of *bona fide* ncRNA sequences by analysing 6559 ncRNAs from nine Model Organism Databases and other curated resources, including DictyBase[40], FlyBase[41], MGI[42], PomBase[43], SGD[44], TAIR[45], WormBase[46], HGNC[47] and EcoCyc[48]. These sequences represent a wide taxonomic distribution, including bacteria (*E. coli*), fungi (*Saccharomyces cerevisiae* and *Schizosaccharomyces pombe*), lower eukaryotes (*Dictyostelium discoideum*), plants (*Arabidopsis thaliana*), as well as other organisms of general interest, such as fly, worm, mouse, and human. R2DT-generated 2D diagrams for the majority of the selected sequences (5663 diagrams or 86.3%), consistent with the RNA type (rRNA, tRNA, snRNA, snoRNA, SRP RNA) and length (25-10,000 nucleotides) of the sequence dataset.

We classified each nucleotide in the resulting diagrams according to whether it matched a template and found that 90.6% of nucleotides were displayed using the nucleotide locations encoded in the templates, while 6.0% of nucleotides represented insertions compared to the templates, and 3.4% of nucleotides matched the templates but required automatic repositioning by

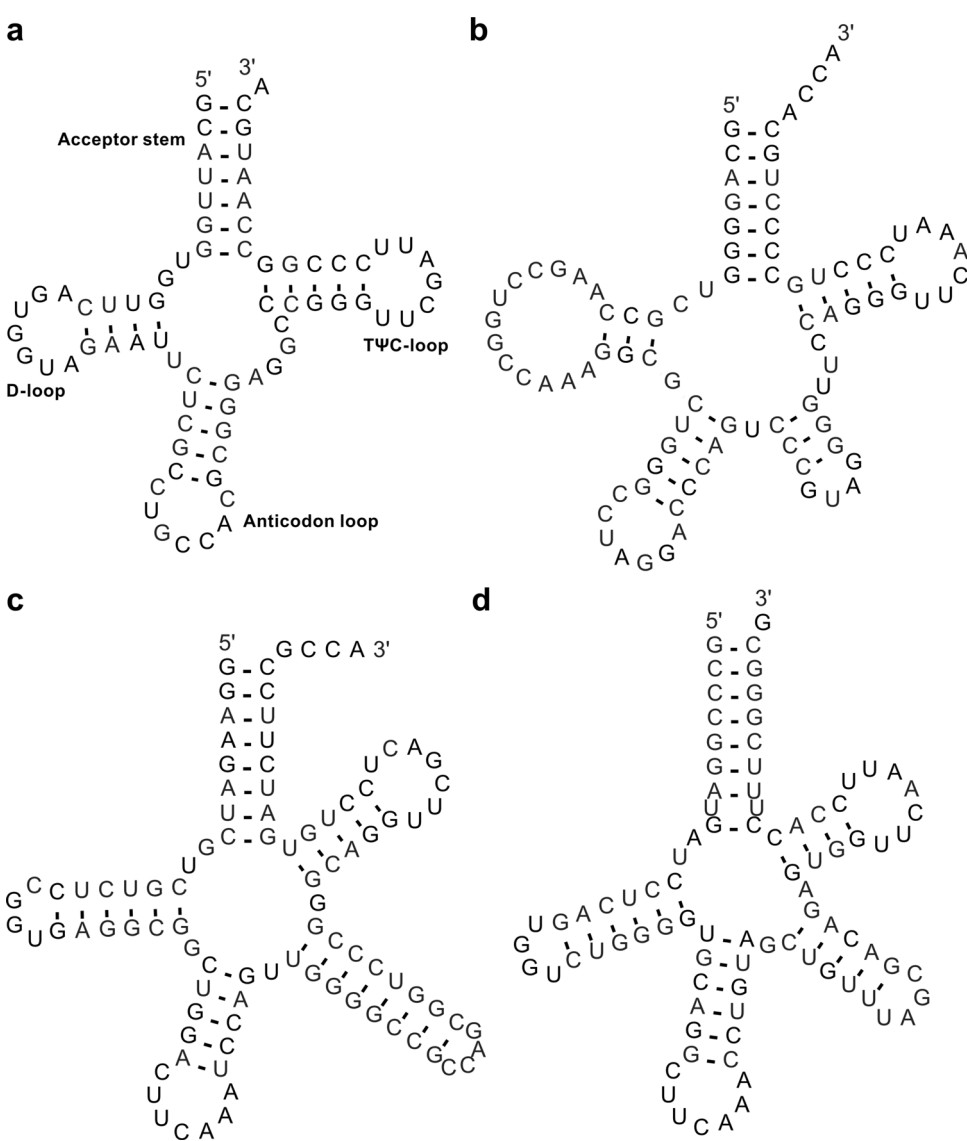

**Fig. 4 Examples of tRNA 2D structure visualisations generated by R2DT. a** Human tRNA-Gly-GCC-2 is an eukaryotic type I tRNA (URS000013B42D_9606). **b** *Methanocaldococcus jannaschii* tRNA-Leu-TAG-1 is an archaeal type II tRNA (URS00001612D2_243232). **c** *Escherichia coli* K-12 tRNA-SeC-TCA-1 (URS00002F9CDD_511145) is a bacterial selenocysteine tRNA with an 8/5 fold[67]. **d** Mouse tRNA-SeC-TCA-1 (URS0000176051_10090) is an eukaryotic selenocysteine tRNA with a 9/4 fold[68].

the Traveler software (Table 2). Overall 94.0% of the nucleotides were visualised using the template coordinates, indicating that the diagram layouts are similar to the corresponding templates. To further confirm the agreement between the templates and the diagrams, we manually inspected 1043 2D diagrams from human and *E. coli* (based on the HGNC and EcoCyC sequences) to identify any modes of failure, such as overlapping structural regions. This process identified only 24 suboptimal diagrams (2.3%) that were characterised by overlapping helices and other artifacts (all diagrams can be seen in Supplementary Data 1), while the remaining 1019 (97.7%) diagrams produced error-free diagrams, indicating a close correspondence between the template and rendered sequence.

To eliminate bias from the use of model organisms (which tend to have the most experimental data), and to also demonstrate the scalability of R2DT, the nucleotide classification analysis was extended to a broad range of sequences from a wide taxonomic distribution by processing all ncRNA sequences from RNAcentral, aiming to test as many realistic use cases as possible. As of release 15 RNAcentral contained 16,107,505 sequences from 896,307 NCBI taxonomic identifiers including ncRNA types not represented by the R2DT template library, such as lncRNA or piRNA, as well as partial sequences. R2DT-generated 13,384,186 2D diagrams (83.1% of the total sequences or 87% of all sequences expected to have a 2D diagram), which can be explored at https://rnacentral.org. Similar to the previous case, 94.7% of nucleotides were drawn in the same position as the templates, while 5.3% were inserted or required recalculation of the 2D layout (Table 2) suggesting that the R2DT template library comprehensively captures the conserved core of most structured RNAs and is suitable for visualising diverse RNA sequences. The agreement between the templates demonstrated in large scale testing on a diverse set sequences from RNAcentral and other sources indicates the broad applicability of R2DT for visualising structured RNAs.

## Discussion

We present a comprehensive framework for the ongoing development of consistent, standardised visualisations of RNA 2D

**Table 2 Analysing the similarity between the R2DT diagrams and the templates.**

| Data source | Number of nucleotides positioned exactly as in template | Number of nucleotides inserted compared to template | Number of nucleotides requiring repositioning | Total number of displayed nucleotides | Number of sequences | Number of diagrams |
|---|---|---|---|---|---|---|
| DictyBase | 9497 (83.1%) | 1188 (10.4%) | 746 (6.5%) | 11,431 | 148 | 123 |
| FlyBase | 35,876 (92.6%) | 1485 (3.8%) | 184 (0.5%) | 38,752 | 458 | 236 |
| MGI | 348,088 (91.6%) | 19,936 (5.2%) | 12,111 (3.2%) | 380,135 | 3166 | 3085 |
| PomBase | 21,498 (85.9%) | 2660 (10.6%) | 878 (3.5%) | 25,036 | 191 | 156 |
| SGD | 26,325 (89.2%) | 2433 (8.2%) | 746 (2.5%) | 29,504 | 188 | 161 |
| TAIR | 46,925 (86.7%) | 3160 (5.8%) | 4,057 (7.5%) | 54,142 | 623 | 483 |
| WormBase | 35,510 (91.7%) | 1614 (4.2%) | 1610 (4.2%) | 38,734 | 639 | 376 |
| HGNC | 135,021 (94.9%) | 2639 (1.9%) | 4685 (3.3%) | 142,345 | 972 | 869 |
| EcoCyc | 44,913 (97.%) | 1036 (2.2%) | 367 (0.8%) | 46,316 | 174 | 174 |
| Total | 703,653 (91.8%) | 36,151 (4.7%) | 25,384 (3.3%) | 766,395 | 6559 | 5663 |
| RNAcentral total | 9,038,893,528 (94.7%) | 261,968,286 (2.7%) | 241,927,491 (2.5%) | 9,542,789,305 | 16,107,505 | 13,384,186 |

The counts indicate the number of nucleotides across all diagrams that match that class, while the percentages indicate the fraction of total displayed nucleotides.

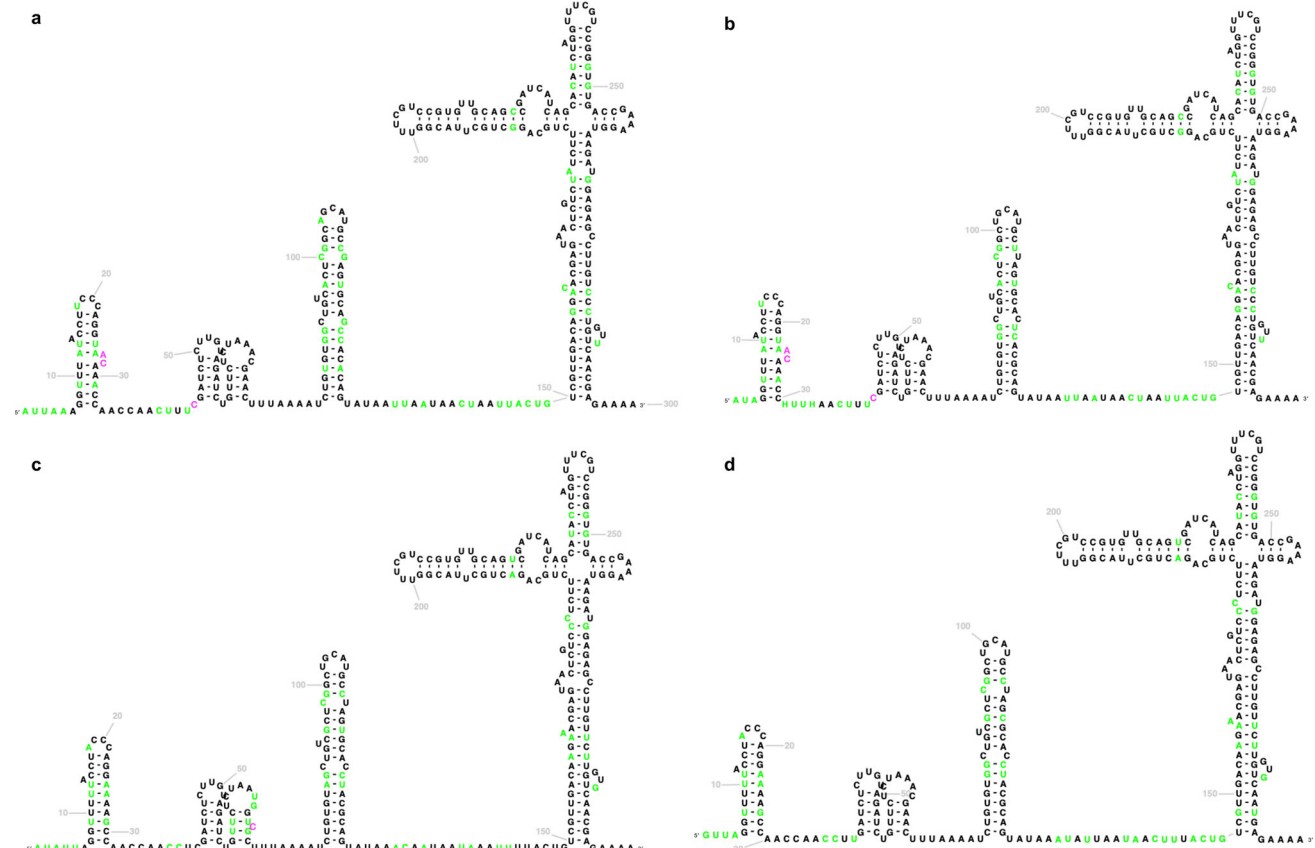

**Fig. 5 Coronavirus 5′ UTR 2D structures displayed using the Sarbecovirus Rfam family (RF03120). a**, **b** SARS-CoV-2 isolates (MT019530.1 and MT263421). **c** SARS Coronavirus Urbani isolate (MK062184.1); **d** Bat SARS Coronavirus HKU3-1 (DQ022305.2). The standardised 2D layouts facilitate structure comparison, with colour coding highlighting the differences between individual sequences and the model. Nucleotides in black are identical to the Rfam consensus sequence and the template, nucleotides shown in green are different between the input sequence and the template, while magenta nucleotides represent insertions.

structures. As additional 2D structure templates are introduced, the pipeline can be extended to cover further RNA types, including structured viral RNAs. For example, as the Coronavirus-specific RNA families were added to the Rfam database in response to the COVID-19 pandemic[49], their 2D structures were included in the template library to enable consistent visualisation of SARS-CoV-2 structured RNAs (Fig. 5), such as the 5′ and 3′

UTRs and frameshifting signal (Rfam accessions RF03120, RF03125, and RF00507, respectively).

The 2D structure diagrams produced by the pipeline represent computational predictions. However, they are based on the accumulated knowledge about the RNA families, as many templates have been curated by experts based on experimental data. The software enables comparative visualisation, as the diagrams encode

a predicted secondary structure and an alignment of a given sequence to its computational model. For example, the diagrams can highlight the structural context of single nucleotide polymorphisms or demonstrate how a member of an Rfam family deviates from the consensus 2D structure (Fig. 5).

While every effort has been made to ensure comprehensive coverage of ncRNA space and the usefulness of the resulting visualisations, R2DT still has some limitations. For example, R2DT cannot generate a diagram if the library does not have a corresponding template or if a sequence matches multiple consecutive templates. In addition, while partial sequences or insertions can be accommodated, some insertions may result in poor visualisations depending on their size and the structural context in which they occur in the template.

R2DT establishes a framework that can be further extended and refined. Importantly, R2DT can be used to generate initial templates that can be manually refined and incorporated into the template library. For example, an rRNA sequence can be submitted to R2DT, the species-specific expansion segment regions can be manually edited, and the resulting diagram can be submitted to R2DT as described above.

In addition, we identified three areas for future development and improvements: (1) expanding and refining the template library. As detailed 2D structures are published, we will integrate them as templates into the R2DT library. In addition, R2DT will benefit from the ongoing development of the Rfam database as additional families are included and additional structural features are annotated in the existing families. (2) Propagating metadata from the templates to the output diagrams. Additional metadata would enable efficient navigation of the 2D structures using the standard numbering schemes for individual nucleotides or structural elements, such as helices and loops (for example, in the rRNAs many structural elements have traditionally assigned numbers, for example, the A-site is located in helix 44). (3) Displaying additional annotations overlaid on top of the 2D diagrams, for example RNA 3D motifs or structure probing reactivities. In addition, the Traveler software already supports pseudoknot visualisation and metadata transfer from the template to the 2D diagrams. These and other improvements will be released on an ongoing basis in the future versions of R2DT. We welcome community feedback and contributions at https://github.com/rnacentral/R2DT/issues.

## Methods
### Constructing the RNA template library
*Covariation-based SSU templates.* The SSU and 5S rRNA templates were downloaded from CRW[2] (http://crw-site.chemistry.gatech.edu/). The 2D structures and templates are based on the comparative analysis of manually curated multiple sequence alignments and are supported by covariation of the interacting base pairs[50]. The 2D structure model diagrams were generated with the Sun Solaris-based version of XRNA[51], manually edited, and written out as both PostScript and PDF files. The R2DT templates have been created based on the CRW bpseq files with the sequence and the 2D structure information, and the PostScript files specifying the position of each nucleotide.

*3D structure-based LSU and SSU templates.* Both LSU and SSU templates have been created using XRNA-GT, an in-house modified version of XRNA software[51], using the pre-existing templates[52] and the manually curated multiple sequence alignments from the SEREB database[53]. The 3D structures were selected using the Representative Sets from RNA 3D Hub[54]. The base-pair interactions in the 3D structures available from the PDB[55] have been annotated using the FR3D software[56]. The 2D layouts were finalised with Adobe Illustrator and written out as SVG files. The final high-quality templates for both LSU and SSU have been integrated in RiboVision[57] and are available at http://apollo.chemistry.gatech.edu/RibosomeGallery.

*tRNA 2D structure templates.* Isotype-specific consensus tRNA sequences and 2D structures were generated using R-scape[58] from the alignments that were used to train and build the corresponding covariance models in tRNAscan-SE[24]. Alignments for training the domain-specific covariance models were split into two

subsets: (1) type I tRNAs (all except type), and (2) type II tRNAs (leucine, serine in bacteria, archaea and eukaryotes, and tyrosine in bacteria). The bacterial tRNA alignments were further filtered to include only one representative tRNA with the same anticodon in each genus due to the original extra large training set (over 73,000 tRNAs). Consensus sequences and the 2D structures of type I and II tRNAs for each domain were then generated using R-scape[58] as the isotype-specific ones. R2R[13] was used for the initial image creation using consensus sequence. Custom adjustments were then made to convert the positions of the images into typical tRNA cloverleaf structure orientation. The templates correspond to tRNAscan-SE 2.0 covariance models that are used to score input sequences against each isotype-specific set and pick the highest scoring domain/template combination. The pseudogene tRNAs, as identified by tRNAscan-SE 2.0, are not currently visualised.

*RNAse P templates.* The majority of the RNAse P 2D layouts (7 archaeal, 7 bacterial, and 1 eukaryotic) have been extracted from the RNAse P database[59]. In addition, several templates have been derived from the RNAse P 3D structures available from the PDB[60], including *Methanocaldococcus jannaschii* (PDB:6K0A), *Bacillus stearothermophilus* (PDB:2A64), *Thermotoga maritima* (PDB:3Q1R), and *Homo sapiens* (PDB:6AHR). The base pairing information for the 3D derived templates was extracted using DSSR software[61], and the 2D layouts were finalised using XRNA-GT.

*Rfam 2D structure templates.* For RNA families without a standard, community-accepted 2D structure layout, we adopted the Rfam consensus 2D structures displayed using the R-scape[58] and R2R[13] software. The R2R software uses a set of rules that lead to consistent diagrams with the standard position of the 5′ and 3′ ends of the sequence. We excluded the lncRNA Rfam families, as well as families that are better represented by specialised templates (for example, the tRNA Rfam families are omitted as the GtRNAdb templates are better suited in this case). The 2675 Rfam templates represent a wide range of RNA types, including microRNAs, snoRNAs, riboswitches, RNA thermometers, IRES RNA, bacterial sRNAs, leaders, and other RNAs from both genomic and metagenomic sources.

### Selecting templates using Ribovore
. The Ribovore software package includes the Infernal software package that implements methods for covariance model- and profile hidden Markov model (HMM)-based analysis of RNA sequences[23]. Ribovore's role in R2DT is to determine the best-matching template model for each input sequence and to validate that the similarity between the sequence and its best-matching model extends across the full length of the sequence. This is achieved by the ribotyper.pl script of the Ribovore package which executes two rounds of Infernal's cmsearch programme. The first round identifies the best-matching model for each sequence by running cmsearch with command-line options "--F1 0.02 --doF1b 0.02 --F1b 0.02 --F2 0.001 --F3 0.00001 --trmF3 --nohmmonly --notrunc --noali". These options run cmsearch in an accelerated mode that computes sequence-only based scores using a profile HMM (ignoring 2D structure), by executing only the first three stages of the HMMER3 profile HMM filter pipeline[62,63]. These first three stages efficiently compute the score of each sequence, but not model alignment boundary positions or accurate sequence alignment boundary positions but these are irrelevant at this step. The model that gives the highest score is selected as the best-scoring template model.

Each sequence's best-matching model is used in the second round of cmsearch, executed with the "--hmmonly" option, that again uses a profile HMM to score sequence only, but this time executing the full HMMER3 filter pipeline such that accurate hit boundaries in sequence and model coordinates are reported. While the second round of cmsearch is slower per model/sequence comparison than the first, only one model is compared to each sequence instead of all models. If the second cmsearch round identifies that there are multiple hits to the model, this indicates that at least some of the input sequence (the intervening sequence between adjacent hits) is either inserted relative to the model, or dissimilar from the expected homologous model region. In this case, the sequence is not evaluated further and no structure diagram will be drawn for the sequence.

Typically, profile HMMs and covariance models are built from multiple sequence alignments, but the SSU and LSU rRNA models used in R2DT were built from the single sequence templates. R2DT uses the Rfam covariance models built from the Rfam seed alignments. If, for a given sequence, the first round of ribotyper.pl cmsearch results in zero models with a score above 20 bits indicating that no significant similarity has been detected to any models, then the second cmsearch round is skipped and the sequence will be analysed in a subsequent step by tRNAscan-SE 2.0 to identify possible similarity against the tRNA models. If no significant hits can be found by either Ribovore or tRNAscan-SE 2.0, R2DT assumes that the sequence does not match any of the templates and will not attempt to draw a structure. Template selection by Ribovore, specifically the first cmsearch round in ribotyper.pl, is typically the slowest step in the R2DT pipeline (the example run times can be seen in Supplementary Table 3) and we hope to speed it up in future versions of R2DT.

### Visualising 2D structures using Traveler
. To produce a layout for an input (target) structure, the Traveler software[25] requires the target and template 2D structures accompanied by the template layout. Both the target and template

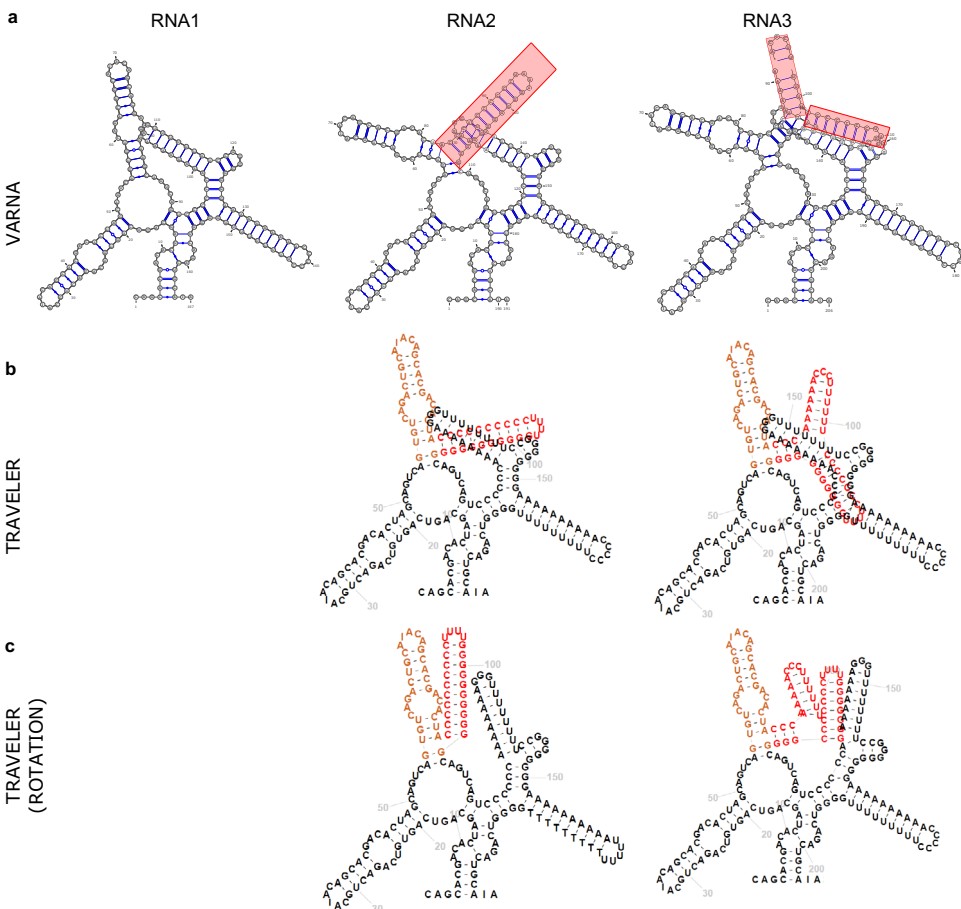

**Fig. 6 Examples of handling overlapping secondary structure elements with the Traveler software.** Sequences RNA2 and RNA3 contain insertions (shown in orange and red) with respect to sequence RNA1. **a** The 2D layouts generated by VARNA (v 3.9); **b** The layouts for RNA2 and RNA3 generated by Traveler when using RNA1 as a template; **c** Same as **b** using Traveler overlap detection and correction. All sequences have been artificially generated to highlight issues with overlapping secondary structure elements.

structures are turned into a tree-based representation, then, a minimum mapping between the trees is found and the template layout is modified based on this mapping to fit the target structure. To support the R2DT pipeline, two major modifications were made to the Traveler software: (i) the ability to provide custom mapping and (ii) optimised hairpin rotation.

Since the target 2D structure is generated by Infernal within the R2DT pipeline, the target-template structure mapping is already known and the original Traveler's mapping procedure is not needed. Therefore, Traveler was modified to use the Infernal-informed tree mapping instead of the original mapping procedure.

Although in most cases the resulting layout is overlap-free, sometimes the target and template differ in such a way that it is not easily possible to fit the target-specific portions of the structure into the template. Therefore, an overlap detection process was implemented in Traveler allowing to rotate the overlapping parts of the structure so that the number of overlaps is minimised. Specifically, Traveler detects the hairpin segments and checks intersection with the rest of the structure. In the case of non-empty overlap, all 30° rotations of the hairpin are tested and the one with the lowest number of overlaps is accepted. As rotations of a single hairpin can open space for further improvements, the process is repeated several times to further decrease the number of overlaps.

While the procedure described above is conservative, a more aggressive approach would lead to target layouts with substantial deviations from the template layout, thus defeating the purpose of template-based layout. An example of a limitation of this approach, where the insertion could not be correctly accommodated, is shown in Fig. 6. Here, RNA3 is oriented the same way as RNA1 (template), but for the price of intersection of the inserted hairpins. Obviously, the larger the difference between the template and target structure, the more pronounced the issue is.

**Pipeline implementation.** The R2DT software is implemented in Python and is packaged using containers to create pre-configured, reproducible environments that support Docker and Singularity platforms. The software has been deployed

within the EMBL-EBI Job Dispatcher framework[64] that provides a web API for submitting jobs and retrieving the results (https://www.ebi.ac.uk/Tools/common/tools/help). The results are visualised with a reusable web component implemented in React that can be embedded into any website (https://github.com/RNAcentral/r2dt-web).

**Reporting summary.** Further information on research design is available in the Nature Research Reporting Summary linked to this article.

## Data availability
The set of precomputed RNAcentral 2D structures are available at https://rnacentral.org. The diagrams are continuously updated as additional templates are developed or algorithm improvements are made. The sequences used in this study were obtained from RNAcentral release 15 under accession codes URS000080E226_274; URS0000ABD82A_9606; URS00000F9D45_9606; URS000075B5A7_9606; URS000013F331_9606; URS000026BDF0_9606; URS0000149178_9606; URS0000051AF4_562; URS000013B42D_9606; URS00001612D2_243232; URS00002F9CDD_511145; URS0000176051_10090, and EcoCyc release 24.0.

## Code availability
The R2DT source code (v1.1) and documentation are available on GitHub under the Apache 2.0 License (https://github.com/rnacentral/R2DT)[65]. An R2DT web server can be found at https://rnacentral.org/r2dt and its source code is available at https://github.com/RNAcentral/r2dt-web. A custom version of XRNA-GT (v1.1) is available at https://github.com/LDWLab/XRNA-GT[66].

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

## Acknowledgements

The authors would like to thank the RNAcentral Consortium for contributing data to RNAcentral as well as the organisers of the 2018 Benasque RNA meeting where this project originated. The authors also thank Mark Ditzler (NASA Ames Research Center) for assistance with preparation of the 3D-based RNAse P templates. This work was supported by Biotechnology and Biological Sciences Research Council (BBSRC) [BB/N019199/1], and by the Intramural Research Program of the National Library of Medicine at the NIH. This work was supported by NASA [80NSSC18K1139] (L.D.W. and A.S.P.).This research was funded in whole, or in part, by the Wellcome Trust [218302/Z/19/Z]. For the purpose of Open Access, the author has applied a CC BY public copyright licence to any Author Accepted Manuscript version arising from this submission.

## Author contributions

B.A.S. generated the diagrams for RNAcentral sequences, performed validation, contributed code, and wrote the manuscript. D.H. adapted the Traveler software to the needs of the project and wrote the manuscript. E.P.N. contributed code, helped with the Ribovore and Infernal software, and wrote the manuscript. CER developed the R2DT web server. F.M. implemented the R2DT API. J.J.C. and R.G. provided the covariation-based SSU and 5S templates. ASP produced the 3D structure-based LSU and SSU templates and the RNAse P templates. A.M. produced the LSU templates. C.M. revised the XRNA-GT code and produced the LSU templates. A.S.P. and L.D.W. coordinated the Georgia Tech team and wrote the manuscript. P.P.C. and T.M.L. produced the tRNA templates, helped with the tRNAscan-SE 2.0 software, and wrote the manuscript. RDF coordinated the project and wrote the manuscript. A.I.P. conceived and implemented the R2DT software, wrote the manuscript, and coordinated the project.

## Funding

## Competing interests

The authors declare no competing interests.
