## [Peer Review File · Nature Communications]

Reviewers' Comments:

Reviewer #1:

Remarks to the Author:

The R2DT framework described here presents a highly automated way to draw the secondary structures of RNAs from known families based on a pre-defined templates. Currently, there are various layout algorithms for secondary structures in use, which can yield drawings of the same RNA structure that appear completely different. Even when using the same layout algorithm drawings of closely related RNAs can be visually different. Template based drawing methods offer a solution to this problem, as they allow to define a standard layout for a family of RNAs, which in turn facilitates discussing structural properties of RNA.

The perhaps most impressive part of the work is the huge collection of templates provided with R2DT. The template library not only covers most RFam families, but also provides layouts for many subfamilies of ribosomal RNAs, including some based on measured tertiary structures complete with expansion elements. There is even the possibility of further community driven expansion of the template library, such that newly detected ncRNA families can now be published together with their standard layout.

The second important point is the template choice. For this purpose, each template is associated with a covariance model. The alignment of the query sequence to the CM is then used both to select the optimal template and to define the mapping between query and template. The procedure is state of the art and also represents an improvement on the original Traveler algorithm, which uses a tree alignment between the structures of the template and the query RNA.

The pipeline is fully automatic and thus ideal for use in web servers such as RNAcentral. A minor criticism remains, that it is unclear how the user can deviate from the automated layout. It should be easy, for example, to manually choose a template and perhaps even provide a handcrafted alignment of template and query sequence. Moreover, while the manuscript talks about "folding" the RNA using cmalign, one should be aware that this only inserts base pairs that are part of the model, and thus regions without structure conservation will remain single stranded. Again it is not clear whether and how the user can use their own structure (be it predicted or measured), apart from building a new template.

The software itself comes with a number of examples, but needs more proper documentation beyond what can be found in the Readme file..

Reviewer #2:

Remarks to the Author:

In this paper Sweeney and colleagues introduced R2DT, a tool for the template-based visualization of RNA structures. The tool is definitely an important resource, and I am impressed by the amount of work the authors carried out. I have a few improvements I would recommend to the authors to make their work more readily usable in the RNA community:

- Besides the included templates, I would suggest to include some additional templates that are for sure available, but that I seem to have missed in the authors-provided list, for example, RNases P and MRP
- It is not entirely clear how new templates can be created from scratch and included in the database for automatic selection
- An important feature would be the possibility to add annotations such as boxing around significantly

covarying base-pairs, or structure probing reactivities. It is not clear whether these features can be readily annotated in an automatic fashion by R2DT

- What happens in case the tool does not find an optimal template for the provided RNA? Or whether the automatically-selected template only matches part of the provided RNA? Is it possible for the program to also "attempt" to plot the structure even in the absence of a template? I am thinking of well known algorithms such as the NAView, or the loop-resolution approach used by R2R.

Reviewer #3:

Remarks to the Author:

The authors describe in their manuscript "R2DT: computational framework for template-based RNA secondary structure visualisation across non-coding RNA types" a framework for the generation of consistent and stable RNA secondary structures and their drawings. In order to achieve this goal, first the RNA type of the sequence is detected. Then the RNA sequence is folded and drawn according to the template for this RNA type. The drawings shown in the manuscript are visually well readable and stable against slight changes in the sequence. The framework was tested against the entire database of RNACentral and shows satisfactory results. Nevertheless, the manuscript is not ready for publication and several important issues need to be resolved, that are detailed next.

Major Issues:

1) The authors claim in their manuscript that no RNA drawing algorithm exists so far that produces stable and consistent drawings for similar RNAs. The paper "RNApuzzler: efficient outerplanar drawing of RNA-secondary structures" by Wiegrefe et al. [1] shows that their algorithm also produces stable and consistent RNA drawings even without the usage of templates. The authors demonstrate this in their manuscript also with SSU RNAs, like the authors of this manuscript. Therefore, I see the contribution of this manuscript in the usage of the created templates, which allow different RNA drawing styles depending on the RNA type, rather than in the stability of stable drawings which is provided by the aforementioned publications as well as by others.

2) The framework described by the authors is not a pure drawing algorithm. In addition, the RNAs to be drawn are also pre-folded using a folding template. This leads to as few deviations as possible from the desired layout and the drawing is as stable as possible. Although this procedure is legitimate and seems to make sense, it seems, that most of the stability is due to the constraint folding of the RNAs and not to the subsequent drawing algorithm. Moreover, other drawing algorithms produce similar drawings with similar input data. I would appreciate a more detailed and precise discussion of this point in the manuscript.

3) Due to Concern 2) the title of the manuscript is also misleading. I recommend to change it.

4) The Related Work Chapter is by far the weakest part of the manuscript. The authors describe only very rudimentary the state-of-the-art in drawing

RNA secondary structures and do not discuss other approaches for the comparative comparison of RNA structures. The statement that most drawing algorithms are based on force directed layouts is not correct. The cited tools and algorithms VARNA, RNAView, 3DNA, PseudoViewer, and R2R do not use FDL-based algorithms to the best of my knowledge. Moreover, the statement "None of these methods can produce useful diagrams for large RNA structures, such as the small and large subunit ribosomal RNAs (SSU and LSU, respectively" (line 59-60) is not correct as described in Issue number 1) before.

Further, there is no in-depth discussion of the newer literature [1], [2] and a standard algorithm [3], which is frequently used, in this literature review. The state-of-the-art reports of Wiese et al. [4] and Ponty et al. [5] on the subject of RNA visualization are also not cited.

Finally, the comparative representations of secondary structures by using dotplots [6-8] and arcplots [9-10] are entirely missing. Only the cited R2R follows a similar consensus approach as R2DT, but this is not discussed further in the manuscript by the authors.

I recommend a comprehensive revision of the section.

5) According to the description, the used Traveler algorithm can modify the positions of individual bases in the sequence so that they fit better to the template. This procedure is questionable from a visualization point of view, because it also changes the interpretation of the drawings by the expert. I would recommend a more detailed discussion of this procedure.

6) The authors evaluate the algorithm for different characteristics in detail, but it is not clear how much computing time the algorithm needs. Own tests (Intel i7-9750H CPU, 64GB RAM, Docker Image) showed that the computing time is quite high due to the template matching. The provided website shows similar long computing times. I recommend the authors to analyze the computation time and compare it to those of other algorithms.

7) The presented intersection detection of Traveler is simple and does not always converge. There are intersections that cannot be corrected by rotating only Hairpin Loop segments. I would like to have a more detailed discussion of the limitations of this approach as well as the effects on the template representation in the manuscript.

8) The authors describe in their validation section, that a small part of RNA structures cannot be drawn with R2DT. Only further on in the methods section the authors describe the reason for this behavior, since the template cannot be identified uniquely. I recommend that this limitation of the framework is explained in the manuscript before the validation section, otherwise this limitation seems unclear until the methods section.

Minor Issues:

1) Figure 1: The image shown in d) was not generated by FORNA. Probably the visualizations in d) and e) are swapped.

2) Figure 2 appears redundant to the text and offers little information.

3) Line 179-180 "The single page LSU layouts enable R2DT to visualise the LSU 2D structures automatically, which has not been possible until now"

With the new structure prediction, however, all other algorithms can also draw this structure as a single page layout. Therefore I would recommend to weaken this statement.

4) Unfortunately, the used RNAs do not have a unique IDs in the manuscript, so that the drawings and foldings are difficult to reproduce. I recommend to provide unique IDs for the RNA sequences used.

5) The layout of the bibliography is inconsistent.

6) One has to build the Docker Image by oneself, because there is no further documentation on how to start the pre-built Docker Image.

Due to all these issues, I recommend an extensive revision of the manuscript based on my recommendations in a major revision.

[1] Wiegrefe, Daniel, et al. "RNApuzzler: efficient outerplanar drawing of RNA-secondary structures." *Bioinformatics* 35.8 (2019): 1342-1349.

[2] Shabash, Boris, and Kay C. Wiese. "jViz. RNA 4.0—Visualizing pseudoknots and RNA editing employing compressed tree graphs." *Plos one* 14.5 (2019): e0210281.

[3] Brucoleri, Robert E., and Gerhard Heinrich. "An improved algorithm for nucleic acid secondary structure display." *Bioinformatics* 4.1 (1988): 167-173.

[4] Shabash, Boris, and Kay C. Wiese. "RNA Visualization: Relevance and the Current State-of-the-art Focusing on Pseudoknots." *IEEE/ACM Transactions on computational Biology and Bioinformatics* 14.3 (2016): 696-712.

[5] Ponty, Yann, and Fabrice Leclerc. "Drawing and editing the secondary structure (s) of RNA." *RNA Bioinformatics*. Humana Press, New York, NY, 2015. 63-100.

[6] Charif, Delphine, and Jean R. Lobry. "SeqinR 1.0-2: a contributed package to the R project for statistical computing devoted to biological sequences retrieval and analysis." *Structural approaches to sequence evolution*. Springer, Berlin, Heidelberg, 2007. 207-232.

[7] Daniel Gerighausen, Alrik Hausdorf, Sebastian Zänker, Dirk Zeckzer (2017). iDotter - an interactive dot plot viewer. In 25th International Conference in Central Europe on Computer Graphics, Visualization and Computer Vision 2017.

[8] Gruber, Andreas R., Stephan H. Bernhart, and Ronny Lorenz. "The

ViennaRNA web services." RNA bioinformatics. Humana Press, New York, NY, 2015. 307-326.

[9] Wattenberg, Martin. "Arc diagrams: Visualizing structure in strings." IEEE Symposium on Information Visualization, 2002. INFOVIS 2002.. IEEE, 2002.

[10] Lai, Daniel, et al. "R-CHIE: a web server and R package for visualizing RNA secondary structures." Nucleic acids research 40.12 (2012): e95-e95.

Response to Reviewers

Reviewer #1 (Expertise: RNA structural prediction):

The R2DT framework described here presents a highly automated way to draw the secondary structures of RNAs from known families based on pre-defined templates. Currently, there are various layout algorithms for secondary structures in use, which can yield drawings of the same RNA structure that appear completely different. Even when using the same layout algorithm drawings of closely related RNAs can be visually different. Template based drawing methods offer a solution to this problem, as they allow to define a standard layout for a family of RNAs, which in turn facilitates discussing structural properties of RNA.

The perhaps most impressive part of the work is the huge collection of templates provided with R2DT. The template library not only covers most RFam families, but also provides layouts for many subfamilies of ribosomal RNAs, including some based on measured tertiary structures complete with expansion elements. There is even the possibility of further community driven expansion of the template library, such that newly detected ncRNA families can now be published together with their standard layout.

The second important point is the template choice. For this purpose, each template is associated with a covariance model. The alignment of the query sequence to the CM is then used both to select the optimal template and to define the mapping between query and template. The procedure is state of the art and also represents an improvement on the original Traveler algorithm, which uses a tree alignment between the structures of the template and the query RNA.

The pipeline is fully automatic and thus ideal for use in web servers such as RNACentral. A minor criticism remains, that it is unclear how the user can deviate from the automated layout. It should be easy, for example, to manually choose a template and perhaps even provide a handcrafted alignment of template and query sequence.

We agree with the Reviewer that in some cases it could be useful to be able to bypass the automatic template selection, so a new advanced option was implemented in the R2DT web server allowing the user to choose a template from a searchable dropdown list (see <https://rnacentral.org/r2dt>). The corresponding functionality is also available in the standalone software through two new command line options for listing all templates and specifying a template to be used for visualisation. The manuscript has been updated to describe this functionality and refer to the latest version of R2DT (v1.1).

Moreover, while the manuscript talks about "folding" the RNA using calign, one should be aware that this only inserts base pairs that are part of the model, and thus regions without structure conservation will remain single stranded.

This is indeed an important point that is addressed in the section "Automatic pipeline for template selection and 2D structure visualisation" which reads as follows: "It is important to

note that R2DT does not attempt to fold the unstructured regions found in some templates or predict the structure of the insertions relative to the template.”

Again it is not clear whether and how the user can use their own structure (be it predicted or measured), apart from building a new template.

As the Reviewer suggests, the recommended way to use a structure that is not part of the R2DT template library is to build a new template. We clarified this in the manuscript in the section “Community expansion of the 2D template library”. This process is documented at <https://github.com/RNAcentral/R2DT#how-to-add-new-templates>.

The software itself comes with a number of examples, but needs more proper documentation beyond what can be found in the Readme file.

We significantly expanded the documentation with detailed installation and usage instructions, including a new option for manually selecting a template (see <https://github.com/rnacentral/r2dt/>).

Reviewer #2 (Expertise: RNA biology and structural prediction):

In this paper Sweeney and colleagues introduced R2DT, a tool for the template-based visualization of RNA structures. The tool is definitely an important resource, and I am impressed by the amount of work the authors carried out. I have a few improvements I would recommend to the authors to make their work more readily usable in the RNA community:

- Besides the included templates, I would suggest to include some additional templates that are for sure available, but that I seem to have missed in the authors-provided list, for example, RNases P and MRP

The RNase P and MRP RNAs have initially been represented in the R2DT template library with seven Rfam-based templates. Following the Reviewer’s suggestion, we created 15 new RNase P templates based on the RNase P database created by James Brown (PMID:7524025). Although the database is no longer online, we were able to locate some of the original RNase P images and convert them into R2DT templates. In addition, we included four newly-curated 3D structure-based templates (Methanocaldococcus jannaschii, Bacillus stearothermophilus, Thermotoga maritima, and Homo sapiens). Together, these 19 templates replaced four out of seven Rfam-based templates (Nuclear RNase P, Bacterial RNase P classes A and B, and Archaeal RNase P). The three remaining Rfam-based templates (MRP, Plasmodium RNase P, and RNase P truncated form) will be updated in a future R2DT release. The total number of templates increased from 3,632 to 3,647 (Table 1 and the rest of the manuscript have been updated to reflect the increase). A discussion of the new RNase P templates has been included in the manuscript and a human RNase P structure has been added to Figure 1, replacing MoCo riboswitch in panel e.

- It is not entirely clear how new templates can be created from scratch and included in the database for automatic selection

The creation of new templates is described in manuscript under the section entitled “Community expansion of the 2D template library” and is also documented on GitHub (<https://github.com/RNAcentral/R2DT#how-to-add-new-templates>). To summarise, a bespoke version of the XRNA software (<https://github.com/LDWLab/XRNA-GT>) can be used to import the R2DT-generated SVG files and adjust the 2D layouts (for example, by changing the orientation of RNA helices or edit base pairs). XRNA-GT can also export the files required for the creation of the R2DT templates. This is further supported by a newly added R2DT feature allowing users to manually select a closely-related template to get a draft template before adjusting it with XRNA-GT. The XRNA-GT workflow has been successfully used internally to produce the 3D-based SSU templates as well as the RNase P templates described above. In addition, the R2DT documentation contains example files in bpseq, fasta, and xml formats that can be converted to R2DT templates using example scripts.

- An important feature would be the possibility to add annotations such as boxing around significantly covarying base-pairs, or structure probing reactivities. It is not clear whether these features can be readily annotated in an automatic fashion by R2DT

We agree with the Reviewer that this would be a very useful feature. We plan to implement it in future versions and have updated the Discussion section accordingly. However, this functionality will take significant time to implement. As the Reviewer acknowledged previously, our work is an important resource for the community and this functionality will be added over time.

- What happens in case the tool does not find an optimal template for the provided RNA? Or whether the automatically-selected template only matches part of the provided RNA? Is it possible for the program to also "attempt" to plot the structure even in the absence of a template? I am thinking of well known algorithms such as the NAView, or the loop-resolution approach used by R2R.

For sequences that do not match any template, R2DT does not produce any output files. In case of partial sequence matches, a partial 2D diagram will be generated (for example, the following 16S rRNA structure is truncated on the 3' end but is still visualised using a SSU template: <https://rnacentral.org/rna/URS0000183ACD/1166018?tab=2d>). We have updated the manuscript to describe R2DT's behaviour if no matching template is found (see section “Automatic pipeline for template selection and 2D structure visualisation”). At this time, we do not plan to draw diagrams without templates as this task can be accomplished with other existing tools.

Reviewer #3 (Expertise: RNA structural prediction):

The authors describe in their manuscript "R2DT: computational framework for template-based RNA secondary structure visualisation across non-coding RNA types" a framework for the generation of consistent and stable RNA secondary structures and their drawings. In order to achieve this goal, first the RNA type of the sequence is detected. Then the RNA sequence is folded and drawn according to the template for this RNA type. The drawings shown in the manuscript are visually well readable and stable against slight changes in the sequence. The framework was tested against the entire database of RNAcentral and shows satisfactory results. Nevertheless, the manuscript is not ready for publication and several important issues need to be resolved, that are detailed next.

Major Issues:

- 1) The authors claim in their manuscript that no RNA drawing algorithm exists so far that produces stable and consistent drawings for similar RNAs. The paper "RNApuzzler: efficient outerplanar drawing of RNA-secondary structures" by Wiegrefe et al. [1] shows that their algorithm also produces stable and consistent RNA drawings even without the usage of templates. The authors demonstrate this in their manuscript also with SSU RNAs, like the authors of this manuscript. Therefore, I see the contribution of this manuscript in the usage of the created templates, which allow different RNA drawing styles depending on the RNA type, rather than in the stability of stable drawings which is provided by the aforementioned publications as well as by others.

We thank the Reviewer for the detailed feedback. While other methods may generate stable drawings, their output diagrams are not guaranteed to be as biologically meaningful. For example, the SSU and LSU rRNA diagrams produced without templates do not reflect the 3D architecture of the ribosome that is captured in the manually curated templates. This point is further illustrated in the following response.

- 2) The framework described by the authors is not a pure drawing algorithm. In addition, the RNAs to be drawn are also pre-folded using a folding template. This leads to as few deviations as possible from the desired layout and the drawing is as stable as possible. Although this procedure is legitimate and seems to make sense, it seems that most of the stability is due to the constraint folding of the RNAs and not to the subsequent drawing algorithm. Moreover, other drawing algorithms produce similar drawings with similar input data. I would appreciate a more detailed and precise discussion of this point in the manuscript.

Although other software could generate similar 2D diagrams following small modifications in the input sequence, this is not guaranteed. For example, consider the following diagrams generated by VARNA (v3.9). The sequence on the right is identical to the one on the left with the exception of an additional hairpin inserted on the 5' end of the structure, which is sufficient to cause significant changes in the layout.

However, if the left structure is used as a template and the right one as a target by the Traveler software, the resulting layout preserves the orientation of the template while accommodating the additional helix.

The entire R2DT pipeline has been designed to ensure that the template structures are reproduced as faithfully as possible.

- 3) Due to concern 2) the title of the manuscript is also misleading. I recommend changing it.

As far as we understand, the Reviewer is concerned that the users may not be aware that R2DT not only draws a secondary structure image but also predicts the secondary structure of the input sequence to compare it with the template. However, we do not agree with the Reviewer's assessment that the title is misleading, as both the folding and drawing aspects of the software are captured in the title that defines the R2DT's objective as "template-based RNA secondary structure visualisation." We changed the wording in the Discussion section to make it clear that the pipeline involves a secondary structure prediction step.

- 4) The Related Work Chapter is by far the weakest part of the manuscript. The authors describe only very rudimentary the state-of-the-art in drawing RNA secondary structures and do not discuss other approaches for the comparative comparison of RNA structures. The statement that most drawing algorithms are based on force directed layouts is not correct. The cited tools and algorithms VARNA, RNAView, 3DNA, PseudoViewer, and R2R do not use FDL-based algorithms to the best of my knowledge. Moreover, the statement "None of these methods can produce useful diagrams for large RNA structures, such as the small and large subunit ribosomal RNAs (SSU and LSU, respectively)" (line 59-60) is not correct as described in Issue number 1) before. Further, there is no in-depth discussion of the newer literature [1], [2] and a standard algorithm [3], which is frequently used, in this literature review. The state-of-the-art reports of Wiese et al. [4] and Ponty et al. [5] on the subject of RNA visualization are also not cited. Finally, the comparative representations of secondary structures by using dotplots [6-8] and arcplots [9-10] are entirely missing. Only the cited R2R follows a similar consensus approach as R2DT, but this is not discussed further in the manuscript by the authors. I recommend a comprehensive revision of the section.

We would like to thank the review for this constructive feedback. The introduction has been updated following the Reviewer's suggestion. The reference numbering has changed throughout the entire manuscript.

- 5) According to the description, the used Traveler algorithm can modify the positions of individual bases in the sequence so that they fit better to the template. This procedure is questionable from a visualization point of view, because it also changes the interpretation of the drawings by the expert. I would recommend a more detailed discussion of this procedure.

Traveler aims to limit template modifications as much as possible and preserve the general template topology. However, some nucleotides might still need repositioning in order to accommodate the insertions and deletions in the sequence compared to the template. This is described in detail in the original Traveler publication (<https://bmcbioinformatics.biomedcentral.com/articles/10.1186/s12859-017-1885-4>). Below is an example from that paper demonstrating how the structural elements are shifted to accommodate various types of insertions (panel a shows an extension of the stem while in panel b an additional bulge is inserted). Without nucleotide repositioning these insertions could not be incorporated, so the ability to adjust the template layout is essential. However, it is indeed possible that the adjustment could lead to structural overlaps in the diagrams (see the following answer).

- 6) The authors evaluate the algorithm for different characteristics in detail, but it is not clear how much computing time the algorithm needs. Own tests (Intel i7-9750H CPU, 64GB RAM, Docker Image) showed that the computing time is quite high due to the template matching. The provided website shows similar long computing times. I recommend the authors to analyze the computation time and compare it to those of other algorithms.

As R2DT is the only available tool that performs template-based RNA 2D structure visualisation for a comprehensive set of RNAs, we cannot fairly or appropriately benchmark its performance against other RNA 2D visualisation software because these tools have different objectives than R2DT. While it may be possible to draw a ribosomal RNA without a template faster than R2DT, such visualisations are not always useful (see Figure 1 in the manuscript).

The latest version of R2DT (v1.1) is significantly faster than the original version due to multiple performance optimisations. For example, a test suite comprising 22 representative sequences completed ~35% faster than in R2DT v1.0. As the Reviewer pointed out, the performance is limited by the automatic template selection step where a sequence is compared to all available templates. As the template library grows, we will investigate ways of speeding up this process, potentially using more aggressive template clustering and/or hierarchical searching.

It is worth noting that the website performance does not directly reflect R2DT's performance, as the jobs are executed asynchronously using a queuing system. Depending on the number of submitted jobs, the results may be generated slower than on a local machine. However, in some cases the website could outperform a local installation of R2DT as the jobs run on high-specification hardware at the EMBL-EBI data centres. We will monitor the usage of the website resources overtime and make more resources available if the R2DT web interface becomes overwhelmed.

- 7) The presented intersection detection of Traveler is simple and does not always converge. There are intersections that cannot be corrected by rotating only Hairpin Loop segments. I would like to have a more detailed discussion of the limitations of this approach as well as the effects on the template representation in the manuscript.

The overlap detection and repositioning algorithm does not always lead to overlap-free layouts as its purpose is to fix only minor issues. Although it is possible to devise a procedure that would eliminate all overlaps, such a method would likely require more substantial modification of the template to accommodate the indels. That would, in turn, defy the main purpose of the template-based layout which is to have a common topology for a set of sequences visualised with the same template. Therefore we came up with a pragmatic approach which is able to resolve the majority of simple overlaps. The manuscript has been expanded to include this information together with a new figure (Figure 6) showcasing Traveler's abilities and limitations with respect to overlaps.

- 8) The authors describe in their validation section, that a small part of RNA structures cannot be drawn with R2DT. Only further on in the methods section the authors describe the reason for this behavior, since the template cannot be identified uniquely. I recommend that this limitation of the framework is explained in the manuscript before the validation section, otherwise this limitation seems unclear until the methods section.

To address this in the manuscript, the following sentence has been added to the "Automatic pipeline for template selection and 2D structure visualisation" section: If the sequence does not match any templates, the following steps are skipped, and no output files are generated.

Minor Issues:

- 1) Figure 1: The image shown in d) was not generated by FORNA. Probably the visualizations in d) and e) are swapped.

We thank the Reviewer for pointing out this mistake. The Figure 1 legend has been corrected.

- 2) Figure 2 appears redundant to the text and offers little information.

We agree with the Reviewer and have removed Figure 2. The numbering of the remaining figures has been updated accordingly.

- 3) Line 179-180 "The single page LSU layouts enable R2DT to visualise the LSU 2D structures automatically, which has not been possible until now" With the new structure prediction, however, all other algorithms can also draw this structure as a single page layout. Therefore I would recommend weakening this statement.

The sentence has been changed to: "The single page LSU layouts enable R2DT to visualise the LSU 2D structures in standard orientations completely automatically, which has not been possible until now."

- 4) Unfortunately, the used RNAs do not have unique IDs in the manuscript, so that the drawings and foldings are difficult to reproduce. I recommend providing unique IDs

for the RNA sequences used.

We have added the RNACentral unique sequence accessions to figure legends (Figures 1, 3, 4, 5) to unambiguously identify each sequence.

- 5) The layout of the bibliography is inconsistent.

The bibliography has been checked for consistency.

- 6) One has to build the Docker Image by oneself, because there is no further documentation on how to start the pre-built Docker Image.

We expanded the installation and usage instructions to provide detailed examples, including running R2DT using pre-built images from Docker Hub (please see <https://github.com/RNACentral/r2dt>).

Due to all these issues, I recommend an extensive revision of the manuscript based on my recommendations in a major revision.

[1] Wiegrefe, Daniel, et al. "RNApuzzler: efficient outerplanar drawing of RNA-secondary structures." *Bioinformatics* 35.8 (2019): 1342-1349.

[2] Shabash, Boris, and Kay C. Wiese. "jViz. RNA 4.0—Visualizing pseudoknots and RNA editing employing compressed tree graphs." *Plos one* 14.5 (2019): e0210281.

[3] Brucoleri, Robert E., and Gerhard Heinrich. "An improved algorithm for nucleic acid secondary structure display." *Bioinformatics* 4.1 (1988): 167-173.

[4] Shabash, Boris, and Kay C. Wiese. "RNA Visualization: Relevance and the Current State-of-the-art Focusing on Pseudoknots." *IEEE/ACM Transactions on computational Biology and Bioinformatics* 14.3 (2016): 696-712.

[5] Ponty, Yann, and Fabrice Leclerc. "Drawing and editing the secondary structure (s) of RNA." *RNA Bioinformatics*. Humana Press, New York, NY, 2015. 63-100.

[6] Charif, Delphine, and Jean R. Lobry. "SeqinR 1.0-2: a contributed package to the R project for statistical computing devoted to biological sequences retrieval and analysis." *Structural approaches to sequence evolution*. Springer, Berlin, Heidelberg, 2007. 207-232.

[7] Daniel Gerighausen, Alrik Hausdorf, Sebastian Zänker, Dirk Zeckzer (2017). iDotter - an interactive dot plot viewer. In 25th International Conference in Central Europe on Computer Graphics, Visualization and Computer Vision 2017.

[8] Gruber, Andreas R., Stephan H. Bernhart, and Ronny Lorenz. "The ViennaRNA web services." *RNA bioinformatics*. Humana Press, New York, NY, 2015. 307-326.

[9] Wattenberg, Martin. "Arc diagrams: Visualizing structure in strings." *IEEE Symposium on Information Visualization, 2002. INFOVIS 2002.. IEEE, 2002.*

[10] Lai, Daniel, et al. "R-CHIE: a web server and R package for visualizing RNA secondary structures." *Nucleic acids research* 40.12 (2012): e95-e95.

Reviewers' Comments:

Reviewer #1:

Remarks to the Author:

I appreciate the more extensive review of existing tools and especially the addition of options to manually select a template. An option to provide also the alignment of sequence and template could still be useful for expert users (especially given that ribovore runs in HMM mode); I hope this might be added in the future.

I believe the manuscript much improved, but am adding a few minor comments could still improve the final version:

- one point I forgot to mention in the first review, is what happens when more than 1 template is available for a particular RNA family, such as SSU rRNAs with crw and ribosivion templates. If the choice depends purely on alignment score, then you might violate the "similar sequence -> similar layout" rule, when two closely related sequences are assigned to different templates. It might be better to consistently prefer one over the other unless the user specifically chooses a type.

- The new paragraph on existing methods mixes up two aspects of drawing tools: the type of visualization produced (dot plot, arc or circle plot, graph) and algorithm used to generate the layout (force-directed, rule-based).

- Of the five properties, it may not be clear what "modular" means; "have similar appearance" only RNAs from the same family should have similar appearance, more importantly *comparison* should be easy; "3d structure" add "(if known)".

- End of page 24, this may be a good place to mention that when no significant hit can be found by either cmsearch or tRNAscan, R2DT assumes the sequence does not belong to a known family and will not attempt to draw a structure.

With respect to search speed, users will almost always no the organism from which their RNA sequence was taken. This could be used to exclude many templates.

- Are the RNA1, RNA2, RNA3 sequence used in Figure 6 an artificial example?

- The description of the software on github has been slightly improved. I still think it should be more extensive. Especially, a tutorial on adding templates would be helpful if you really hope for community engagement. Eventually, you will need more docu than a single Readme file

Reviewer #2:

Remarks to the Author:

I am content with the authors' responses, and I would like to recommend the article for publication.

Reviewer #3:

Remarks to the Author:

The authors have submitted a comprehensive revision based on the reviewer comments. In doing so, they have addressed the majority of the reviewers' comments. In particular the revision of the Related Work part stands out positively. Also the addition of the limitation of the underlying drawing algorithm to resolve intersections in the drawings is now much clearer.

However, I still think that the title does not reflect the entire work, since constraint folding provides an

important intermediate step for mapping the RNA structure onto the drawing template. Without this step, the entire framework does not work, which is a significant difference from most other RNA drawing algorithms. In my opinion, a good alternative would be "R2DT: computational framework for template-based RNA secondary structure *prediction and* visualisation across non-coding RNA types".

Furthermore, a short benchmark on the performance of the pipeline would also be desirable, even if the results are not comparable to other algorithms. Nevertheless, the results would be informative how the computational time and memory consumption varies between different templates. This could also support statements about the scalability of the framework for larger RNAs.

Response to Reviewers

Reviewer 1

I appreciate the more extensive review of existing tools and especially the addition of options to manually select a template. An option to provide also the alignment of sequence and template could still be useful for expert users (especially given that ribovore runs in HMM mode); I hope this might be added in the future.

We agree with the Reviewer that such a feature could indeed be useful for the expert users who can edit the input alignments instead of tweaking the output diagrams. This option will be implemented in a future version of R2DT, and the progress can be tracked on GitHub (<https://github.com/RNAcentral/R2DT/issues/43>).

I believe the manuscript much improved, but am adding a few minor comments could still improve the final version:

- One point I forgot to mention in the first review, is what happens when more than 1 template is available for a particular RNA family, such as SSU rRNAs with crw and ribovision templates. If the choice depends purely on alignment score, then you might violate the "similar sequence -> similar layout" rule, when two closely related sequences are assigned to different templates. It might be better to consistently prefer one over the other unless the user specifically chooses a type.

R2DT searches the templates in a specific order so that the 3D-based templates are preferentially selected. Specifically, the 3D-based Ribovision templates are searched before the covariation-based CRW templates. The users can override this process by specifying which template or template collection they would like to use on the command line or by manually selecting a template in the web interface (see the section "Automatic pipeline for template selection and 2D structure visualisation" for more details).

- The new paragraph on existing methods mixes up two aspects of drawing tools: the type of visualization produced (dot plot, arc or circle plot, graph) and algorithm used to generate the layout (force-directed, rule-based).

Of the five properties, it may not be clear what "modular" means; "have similar appearance" only RNAs from the same family should have similar appearance, more importantly *comparison* should be easy; "3d structure" add "(if known)".

We have corrected the paragraph as suggested by the reviewer.

- End of page 24, this may be a good place to mention that when no significant hit can be found by either cmsearch or tRNAscan, R2DT assumes the sequence does not belong to a known family and will not attempt to draw a structure.

We thank the reviewer for this comment and have updated the text accordingly.

- With respect to search speed, users will almost always know the organism from which their RNA sequence was taken. This could be used to exclude many templates.

This is a good idea and we will consider it for future versions of R2DT. However, in some cases imposing taxonomic limitations prior to template selection could lead to mistakes (for example, mitochondrial rRNAs are similar to bacterial and not eukaryotic rRNAs).

- Are the RNA1, RNA2, RNA3 sequence used in Figure 6 an artificial example?

This is correct. We have added a clarification to Figure 6 legend.

- The description of the software on github has been slightly improved. I still think it should be more extensive. Especially, a tutorial on adding templates would be helpful if you really hope for community engagement. Eventually, you will need more docu than a single Readme file.

We have continued improving the documentation as suggested by the reviewer and further expanded the section about adding templates. We agree that a dedicated documentation site may become necessary as the project grows and we look forward to engaging with the RNA community.

Reviewer 2

I am content with the authors' responses, and I would like to recommend the article for publication.

Reviewer 3

The authors have submitted a comprehensive revision based on the reviewer comments. In doing so, they have addressed the majority of the reviewers' comments. In particular the revision of the Related Work part stands out positively. Also the addition of the limitation of the underlying drawing algorithm to resolve intersections in the drawings is now much clearer.

However, I still think that the title does not reflect the entire work, since constraint folding provides an important intermediate step for mapping the RNA structure onto the drawing template. Without this step, the entire framework does not work, which is a significant difference from most other RNA drawing algorithms. In my opinion, a good alternative would be "R2DT: computational framework for template-based RNA secondary structure *prediction and* visualisation across non-coding RNA types".

Following suggestions by the Editor and the reviewer, we changed the title to “R2DT is a framework for predicting and visualizing RNA secondary structure using templates”.

Furthermore, a short benchmark on the performance of the pipeline would also be desirable, even if the results are not comparable to other algorithms. Nevertheless, the results would be informative how the computational time and memory consumption varies between different templates. This could also support statements about the scalability of the framework for larger RNAs.

We have added a supplementary table to provide users with a summary of the current performance of R2DT. Sample of R2DT run times for a variety of sequences. Measurements were taken on a CentOS machine with 10G of RAM and on a Intel(R) Xeon(R) Gold 6252 CPU at 2.10GHz. The table is reproduced below.

PDB ID	RNAcentral identifier	Sequence length	RNA type	R2DT run time (m:ss)
4V4Q Chain AA	URS00000ABFE9_562	1,542	Escherichia coli SSU rRNA	0:15
4V4Q Chain BB	URS00004B0F34_562	2,904	Escherichia coli LSU rRNA	0:49
5W4K Chain 1w	URS00005AA258_562	76	Escherichia coli tRNA	0:45
6AHR Chain A	URS000013F331_9606	417	Homo sapiens RNase P	0:10
6EK0 Chain L7	URS000002B0D5_9606	120	Homo sapiens 5S rRNA	0:46
4UG0 Chain L5	URS000086853A_9606	5,070	Homo sapiens LSU rRNA	1:47
6G4W Chain 2	URS0000D56C31_9606	1,882	Homo sapiens SSU rRNA	0:15
3J9M Chain A	URS000080E357_9606	1,559	Homo sapiens mitochondrial SSU rRNA	0:41
6RXV Chain C2	URS0000EEACFC_209285	230	Chaetomium thermophilum U3 snoRNA	0:18